# Opinions and Practices Regarding Electronic Cigarette Use among Middle School Students from Rural Areas of Romania

**DOI:** 10.3390/ijerph19127372

**Published:** 2022-06-16

**Authors:** Tania Elena Tudor, Lucia Maria Lotrean

**Affiliations:** Department of Community Medicine, Iuliu Hatieganu University of Medicine and Pharmacy, 400012 Cluj-Napoca, Romania; llotrean@umfcluj.ro

**Keywords:** electronic cigarettes, adolescents, Romania, rural area

## Abstract

Background: The objectives of the study were to assess awareness, opinions, and practices regarding electronic cigarette (e-cigarette) use, as well as factors associated with their use, among middle-school aged students from rural areas of Romania. Methods: The study sample included 748 middle-school students aged 13–14 years from 24 schools from rural areas situated in two counties from Romania, after receiving parental consent. A cross-sectional study using confidential questionnaires which assessed smoking-related behaviors, and also opinions and practices related to e-cigarettes use, was performed in 2019 among the participating middle-schoolers. Results: 96.3% of the middle schoolers have heard about e-cigarettes. A percentage of 72.7% of the smokers, 50.8% of the ex-smokers, and 15.4% of the non-smokers had tried e-cigarettes at least once in their life; 20.3% of the smokers, 4.8% of the ex-smokers, and 4.5% of the non-smokers reported using e-cigarettes in the last month. The results of multivariate logistic regression analysis pointed out that e-cigarette use at least once during lifetime was associated with having friends who tried e-cigarettes, having stronger beliefs that they can help quit smoking and that they are less dangerous than traditional cigarettes. The intention to use e-cigarettes in the next year and smoking behavior were also correlated with e-cigarettes experimentation, while no gender differences were found. Conclusions: The results underline the importance of having prevention programs and interventions concerning e-cigarettes consumption, since e-cigarettes consumption is spread among Romanian adolescents from rural areas.

## 1. Introduction

Electronic cigarettes (e-cigarettes) are battery-powered devices that heat a liquid that typically contains nicotine, flavors, and other chemicals to produce an aerosol that is inhaled into the lungs [1]. E-cigarettes are a psychosocial phenomenon of the 21st century with serious implications for public and individual health. The significant increase in their popularity and use has raised concerns in the healthcare community regarding their effects [2]. Recent studies from the United States reveal other reasons for trying e-cigarettes among adolescents, such as being in trends with this new device, smaller costs, flavors, easy accessibility, lack of information concerning the use of e-cigarettes, and the great promotion of these devices [3].

A recent study suggests that even if the e-cigarette used contains low levels of nicotine, users are exposed to toxic chemicals with carcinogenic effects, due the vaping of the flavors [4].

The World Health Organization underlines that tobacco products and e-cigarettes pose risks to health, and the safest approach is not to consume. Due to the harmful effects on health, according to Global Tobacco Control 2020, e-cigarettes have been banned in many countries, while in others there are restrictions/regulations on their sale, but also regulations related to the concentration and volume of nicotine contained in e-liquids [5,6,7].

Center for Disease Control and Prevention (CDC) and FDA analyzed representative data from the 2020 National Youth Tobacco Survey (NYTS), a cross-sectional, school based, self-administered survey of U.S. middle school and high school students conducted in January–March 2020. The results showed that 19.6% of high school students and 4.7% of middle school students reported current e-cigarette use (e-cigarette use in the past 30 days) [8]. The same survey among U.S. middle school and high school students conducted in January–May 2021 revealed a decrease of current e-cigarette use (11.3% of high school students and 2.8% of middle school students), but an increase of daily users (27.6% among current high school e-cigarette users, and 8.3% among current middle school e-cigarettes users) [9].

A review study of the prevalence of e-cigarette use among adults and young people in Europe found that the prevalence of using e-cigarettes is higher among adolescents and young adults, males and traditional cigarette users [10].

In 2020 a study was published that analyzed e-cigarette and tobacco cigarettes ever-use, their social correlates, and e-liquid use among adolescents aged between 14–17 years from seven European countries; the results showed that adolescents’ e-cigarette ever use varies greatly between European countries [11].

A recent study that analyzed cross-sectional data on students aged between 11 and 17 years old from the latest available Global Youth Tobacco Survey (GYTS) completed in 17 study sites (16 countries and the Federation of Bosnia and Herzegovina) was used to estimate crude and adjusted prevalence of e-cigarette use by socio-demographic factors, and showed that compared to 2014, the age-adjusted prevalence of e-cigarette use more than doubled in Georgia and Italy, and almost doubled in Latvia. Significantly more male students reported e-cigarette use [12].

Even though there are many countries where e-cigarettes are banned, in several European countries, such as Romania, e-cigarettes are legal and highly promoted. Given the addictive and harmful effects of e-cigarettes, it is essential to identify the factors associated with the initiation and use of e-cigarettes among young people to develop appropriate health education programs and policy measures. Nevertheless, in Romania this type of information is limited, while no information is available regarding e-cigarette use among adolescents from rural areas.

A similar study performed in 2013 in Romania, which assessed opinions and practices regarding e-cigarette use among Romanian high-school students from Cluj-Napoca and Sibiu, showed that the prevalence of e-cigarette use at least once during lifetime was 28.9%, while in the previous month, it was 3%. In the same study, 54.3% thought that e-cigarettes are less dangerous, 52.6% were convinced that e-cigarettes could help smokers to quit, and 45.7% thought that e-cigarettes are used only by smokers [13].

In 2017, a study performed among university students in Romania revealed a prevalence of e-cigarette use of 43.7% at least once during lifetime, while in the previous month it was 8.9% [14].

Given that the prevalence of e-cigarette use among adolescents is increasing, and these devices are widespread and accessible, up-to-date data concerning students’ attitudes toward e-cigarette use is useful to initiate prevention programs that involve factors predisposing to the use of e-cigarettes among adolescents.

This study aims to assess awareness, opinions, and practices regarding e-cigarette use among Romanian middle-schoolers from rural areas and to identify correlates of experimentation with e-cigarettes.

## 2. Materials and Methods

### 2.1. Study Sample and Procedure for Data Collection

The study was approved by the Ethics Commission of “Iuliu Hatieganu” Medicine and Pharmacy University, Cluj Napoca, Romania (code 193 from 19 April 2018).

The schools involved in the study were situated in rural areas from two counties of Romania; one situated in the North-western part of the country (Cluj County) and one from the west side of Romania (Arad County).

Principals of twenty-six schools randomly selected from rural areas situated in the selected ccounties were asked if they agree with the participation of their school in a smoking prevention program.

The schools were randomly selected from the list of secondary schools from villages located in Arad County and Cluj County. We selected the schools from the list of secondary schools made public on the website of the school inspectorate from the two counties. Thirteen out of 87 Gymnasium schools located in Arad County and thirteen out of 92 gymnasium schools located in Cluj County were selected. Twenty-four out of twenty-six school principals agreed to participate with their schools in the project and provided the number of the 7th and 8th grade classes that could participate.

Informed consent of parents for the participation of middle school students in the study was also needed. Out of 1172 that were asked to offer informed consent for students’ participation, 825 parents approved their children’s participation in the study.

In autumn 2019, confidential questionnaires were given to each middle schooler whose parents agreed to participate; the medium time of completing the questionnaire was 50 min. On the first page of the questionnaire, the participants were informed about some rules of filling in the questionnaires, that their participation was voluntary, and that the research team would treat their questionnaires confidentially. The teachers were in the classroom during the data collection, but they did not intervene to assure confidentiality. After the middle schoolers completed the questionnaire, they had to put it in an envelope, seal it, and write their names on it. Middle-schoolers who did not want to participate did not return the questionnaire. A confidential approach was used to be able to develop a longitudinal study over one year concerning these issues.

The researchers collected all the envelopes. The questionnaires were filled in by 760 students because there were 65 who were absent on the days of assessment. Of those 760, 7 middle schoolers had mental illnesses or genetics syndromes in which the ability to understand text is affected, and 5 completed only half of the questionnaire, so the research team excluded them from the study, leaving 748 remaining questionnaires. The distribution by gender was 50.6% girls, 49.4% boys.

### 2.2. Instruments for Data Collection

The study used a confidential questionnaire developed for this study based on previous questionnaires developed and tested in several studies from Romania [13,14,15,16]. It included several sections investigating health risk behavior among participating middle-schoolers, with a special focus on smoking behavior and e-cigarette use. For assessment of smoking related knowledge, attitudes, and behavior a questionnaire based on I-Change model which was tested in previous studies in Romania was used [16], while for assessment of e-cigarette related cognition and behavior a questionnaire developed and used previously in Romania was adapted (the previous questionnaire was kept, while adding few other questions regarding the type of e-cigarette used and tobacco cigarettes smoker status when first tried e-cigarettes) [13,14,15]. The present study includes information collected through the confidential questionnaires regarding the following issues:-Socio-demographic characteristics (age, gender, school);-Awareness and beliefs regarding e-cigarette use (several statements were listed, and students had to declare if they agree or not; the possibilities for answer varied from I totally agree to I totally disagree);-Sources of information about e-cigarette use (the possibilities for answer were: internet/commercials/sales points, newspapers/friends/people from the same school year/parents);-Social influences related to e-cigarette use (middle schoolers were asked if their parents/siblings/friends are using e-cigarettes). The possible answers were yes/no/I do not know; for the question concerning the parents and siblings, they had also the possibility to answer “I do not have brother/sister” or “my mother/father is dead”;-Behavior related to e-cigarette use (students were asked if they have used e-cigarettes at least once during their lifetime and in the previous month);-Reasons for using e-cigarettes at least once during their lives among those who had tried e-cigarettes (e cigarettes are less dangerous/to reduce the number of traditional cigarettes/to quit smoking/curiosity/other friends also tried e-cigarettes/parents tried e-cigarettes);-Type of e-cigarettes used (nicotine free/with nicotine and the amount of contained nicotine as well as if they were flavored/unflavored and which flavor was used);-Tobacco cigarette smoker status when first tried e-cigarettes (middle schoolers were asked if they were tobacco cigarette smokers when they first tried e-cigarettes; the possibilities of answer were yes or no);-Intention to use e-cigarettes in the next year (students were asked if they intended to use e-cigarettes in the next year; the possibilities of answer varied from definitely no to definitely yes);-Behavior related to smoking (smoker status was evaluated by a question in which adolescents were asked to choose a statement that best described their behavior related to smoking (the possibilities for answer were: “I smoke at least once a day”, “I smoke at least once a week”, “I smoke at least once a month”, “I smoke less than once a month”, “I smoke from time to time”, “I quit smoking, but I used to smoke at least once a week”, “I quit smoking, but I used to smoke less than once a week”, “I smoked from time to time, but now I don’t smoke”, “I never smoked a cigarette, not even a few smokes”). Based on those answers, middle schoolers were categorized into three categories: smokers were considered middle schoolers who answered “I smoke at least once a day”, “I smoke at least once a week”, “I smoke at least once a month”, “I smoke less than once a month”, “I smoke from time to time”; ex-smokers were considered those who answered: “I quit smoking, but I used to smoke at least once a week”, “I quit smoking, but I used to smoke less than once a week”, “I smoked from time to time, but now I don’t smoke”, and non-smokers were considered those who never tried tobacco cigarettes;-The intention to quit tobacco cigarette smoking in the future was also investigated among smokers, by a question with different possibilities of answer (I do not want to quit/I want to quit smoking in the future month/future six months/future year/future five years/not in the future five years/I don’t smoke); Answers between “I want to quit smoking in the future month, in the future six months or in the next year” were considered “yes”, while answers between “I don’t want to quit smoking, I want to quit in the future five years/not in the future five years” were considered, “no”.

An identification code was used in the database for each questionnaire; researchers replaced the names of students before the data were entered into the database. The questionnaires which were not fully filled were excluded from the study.

### 2.3. Data Analysis

The percentages were reported to middle-schoolers who had heard about e-cigarettes (N = 720).

Opinions, behaviors, and source of information regarding e-cigarette use as well as intention to use e-cigarettes were investigated for the whole sample (N = 720) as well as for smokers (N = 128)/ex-smokers (N = 124) and non-smokers (N = 468).

The percentages related to reasons for trying e-cigarettes and types of e-cigarettes used were calculated for middle-schoolers who declared that they had heard about e-cigarettes, but they also tried it (N = 212, smokers—95, ex-smokers—57, non-smokers—60). The amount of nicotine contained in e-cigarettes was investigated only in middle-schoolers who declared that they had tried e-cigarettes containing nicotine (N = 51, smokers = 32; ex-smokers = 14; non-smokers = 5).

Univariate logistic regression analyses were performed to assess the correlates of trying e-cigarettes at least once, among the whole study sample, and smokers. The dependent variable was coded as 0—never tried e-cigarettes, and 1—have tried at least once during lifetime. The independent variables included were: age, gender, smoking related behavior (0—non-smoker; 1—ex-smoker; 2—smoker); beliefs about e-cigarettes (e-cigarettes could help people quit smoking; e-cigarettes are less dangerous than traditional cigarettes; e-cigarettes are used only by smokers; using e-cigarettes reduces the possibility for smokers to seek another medical method to quit; some teenagers use e-cigarettes without being traditional cigarettes smokers) the possibilities of answer being grouped in two categories (I totally agree/I partially agree vs. I totally disagree/I partially disagree/I do not know); experimentation with e-cigarettes by friends, mother, father, siblings (yes vs. no). The analyses also included the intention to use e-cigarettes in the future; the possibilities of answer were grouped in two categories (definitely yes/probably yes vs. definitely no/probably no/I do not know); intention to quit smoking in the future (yes vs. no) was also included, for smokers only.

Multivariate logistic regression analysis was also performed with the independent variables included in the analysis being those ones which were statistically significant at univariate analyses (forced entry was used). The dependent variable was coded as 0—never tried e-cigarettes, and 1—have tried at least once during lifetime.

Data analysis was made with IBM SPSS Statistics version 20.0 (IBM Corporation, Armonk, NY, USA). Significant results are reported at *p* < 0.05.

## 3. Results

### 3.1. Awareness, Source of Information, Beliefs, and Social Influences Regarding e-Cigarette Use

The sample of this study is represented by middle-school students who had the consent of the parents to participate and fill in the confidential questionnaires (N = 748).

The results show that of to the whole sample (N = 748), 17.8% were smokers, 17.2% were ex-smokers, and 65.0% were non-smokers. The results from the tables were reported to those who declared that they had heard about e-cigarettes (N = 720).

Table 1 shows that the main source of information regarding e-cigarettes is the internet.

In addition, more than 30% believed that e-cigarettes are less dangerous than traditional cigarettes and can help smokers to quit. A quarter of the students who declared that they had heard about e-cigarettes believed that e-cigarettes are used only by smokers (More than 50% of the smokers and ex-smokers, but also 40% of the non-smokers think that some teenagers use e-cigarettes without being tobacco cigarettes smokers.

The majority of the smokers (78.9%), ex-smokers (66.1%), and non-smokers (37.9%) declared that they have friends who tried e-cigarettes; 5.5 % of smokers, 4% of ex-smokers and 2.8 % of non-smokers had mother who tried e-cigarette, while 3.1% of the smokers, 4.8% of the ex-smokers and 4.9% of non-smokers had father who tried e-cigarettes. 11.7% of the smokers, 6.5% of the ex-smokers, and 2.8% of the non-smokers had siblings who tried e-cigarettes.

### 3.2. Behavior and Intention to Use e-Cigarettes

About 30% of the middle schoolers declared that they used e-cigarettes at least once during their lifetime, while 7.4% of the whole study sample used e-cigarettes in the last month.

Regarding tobacco cigarettes smoking behavior, 38.3% of the smokers and 93.5% of the ex-smokers were not smokers when they first tried e-cigarettes.

Table 2 refers not only to adolescents who declared that they had heard about e-cigarettes but also to those who said that they tried smoking e-cigarettes at least once during their lifetime (N = 212, smokers—95, ex-smokers—57, non-smokers—60). The main reason for trying e-cigarettes in this sample is curiosity.

Concerning the type of e-cigarette used (nicotine/nicotine-free), almost a quarter declared that they used e-cigarettes containing nicotine, and almost a quarter declared that they don’t know if the e-cigarettes used contained nicotine or not. Of those who declared that they tried e-cigarettes containing nicotine (N = 51, smokers = 32; ex-smokers = 14; non-smokers = 5), more than half did not know the amount of nicotine contained in the e-cigarettes that they tried.

The majority of middle-schoolers declared that they tried flavored e-cigarettes (see Table 2).

### 3.3. Correlates of Experimentation with Electronic Cigarettes

Table 3 presents the factors associated with e-cigarette experimentation, exposing the results of univariate and multivariate logistic regression reported to the whole sample (N = 720). The results of univariate logistic regression analyses show that among all study samples, older adolescents were more prone to experiment with e-cigarettes, but no gender differences were found. Middle-schoolers who had stronger beliefs that e-cigarettes help quit smoking and they are less dangerous, but also that some adolescents first use electronic cigarettes without being tobacco cigarettes smokers and also those who think that using e-cigarettes reduces the possibility for smokers to use another method to quit were more likely to try e-cigarettes. Students who had friends who experimented with e-cigarettes were more likely to start using e-cigarettes, as were those who had siblings who experimented with e-cigarettes. Students who heard about e-cigarettes from friends were also more likely to try e-cigarettes, while participation in school health education had a protective effect. Smoking behavior was also correlated with trying e-cigarettes. The intention to use e-cigarettes in the next year was also correlated with electronic cigarettes experimentation. 

Multivariate logistic regression analysis was made for the statistically significant variables from the univariate logistic regression analysis. The results showed that among the whole sample, e-cigarette use at least once during a lifetime was associated with having friends who use e-cigarettes, but also with having strong beliefs that e-cigarette can help quit smoking and that they are less dangerous than traditional cigarettes; the intention to use e-cigarettes in the next year and smoking behavior were also correlated with e-cigarettes experimentation. 

Table 4 presents univariate and multivariate logistic regression results reported to the smokers (N = 128). According to univariate logistic regression results among smokers, the correlates of trying electronic cigarettes were: stronger beliefs that electronic cigarettes are less dangerous, help quitting smoking and some teenagers use e-cigarettes without being tobacco cigarettes smokers as well as having friends who tried e-cigarettes. Multivariate logistic regression analysis was made for the variables which were statistically significant in the univariate logistic regression analysis. The results showed that among smokers, e-cigarette experimentation was related to the beliefs that e-cigarettes are less dangerous.

## 4. Discussion

This study is one of the few studies concerning opinions and practices regarding the use of electronic cigarettes made in Romania. There were performed similar studies concerning the use of electronic cigarettes among Romanian university students (in 2013 and 2017) [14,15], and among high school students in Romania (in 2013) [13], but the sample did not include middle-school aged students from rural areas from Romania. Moreover, the present study was performed in 2019, offering more recent data in comparison with previous studies; due to its changing patterns, smoking e-cigarettes among different groups of young people should be periodically assessed.

Our study revealed a high awareness of e-cigarettes among adolescents from rural areas of Romania, where sales and marketing of these products are widespread.

Another study performed between 2017–2018 among university students in five European countries revealed a prevalence of e-cigarette use of 43.7% at least once during lifetime and 1.1% current use. In our study, performed in 2019, the prevalence of e-cigarettes at least once during lifetime was 31.7%, while in the last month was 7.4% [17].

A similar study among secondary and high school students from a socially disadvantaged rural area in Poland showed a prevalence of e-cigarettes use of 22% at least once during lifetime, but not in the previous month, and 27% declared that they used e-cigarettes in the previous month [18].

Concerning the type of e-cigarette used by adolescents, in our study, 33.7% of the smokers, 24.6% of the ex-smokers, and 8.3% of the non-smokers declared that the e-cigarettes used contained nicotine, while one out of five middle-schoolers did not know if the e-cigarette used contained nicotine or not. As we can see in our study, smokers are more likely to use electronic cigarettes containing nicotine.

The nicotine concentration of e-liquids used by adolescent e-cigarette users is also an important factor involved in the potential harmful effects of e-cigarettes. A cross-sectional survey conducted in 2014 among high-school and middle school students revealed that 37.4% of adolescents involved in the study used nicotine e-liquid, while 34.1% did not know the nicotine concentration of e-cigarette used; 24.1% from our study used e-cigarettes containing nicotine, and 60.8% of them did not know the nicotine concentration containing in the e-liquid used [19]. The fact that in both studies, many adolescents were unaware of the nicotine concentration in the used e-liquid raises concerns about inadvertent nicotine exposure among adolescents.

On the other hand, the majority of the participants in our study used flavored electronic cigarettes. In 2018 a comparative study among adolescent e-cigarette users from five Connecticut high schools and adult e-cigarette users was published, which examined preferences for e-liquid flavors and the total number of flavors preferred between samples of adolescent and adult e-cigarette users, and if these preferences were associated with e-cigarette use frequency for adolescents or adults; the results showed that flavor preferences might play an important role in adolescent e-cigarette use [20].

In our study, there were no found gender differences, while in other studies from Romania, e-cigarettes use was more frequent among male students. Gender differences were also found in a study among secondary and high school students from a socially disadvantaged rural area in Poland, which revealed a higher prevalence of using e-cigarettes among boys. The same study showed that e-cigarette use was widespread among adolescents with smoking parents and friends [18].

In our study, 50% of the whole sample had friends who tried e-cigarettes, less than 5% had parents who tried e-cigarettes, and 5% of the whole sample had siblings who tried e-cigarettes. As in the study performed among Romanian university students in 2013, friends’ influences were correlated with the experimentation of e-cigarettes, but in addition, in this study, the siblings’ influences were correlated with experimentation with e-cigarettes; in the study from 2013 performed on high school students, experimentation with e-cigarettes was associated with parent’s use of e-cigarettes.

On the other hand, a study performed in Romania in 2013, which assessed opinions and practices regarding e-cigarette use among Romanian high-school students, revealed that the main sources of information about e-cigarettes were friends (65.2%), the internet (57.6%), and people from the same school year (39.7%); in our study, besides internet and friends, commercial/sales points are also part of the first three sources of information preferred. In the same study, 54.3% thought that e-cigarettes are less dangerous, 52.6% were convinced that e-cigarettes could help smokers to quit, and 45.7% thought that e-cigarettes are used only by smokers. In our study, the middle-schoolers are less convinced that e-cigarettes are less dangerous, but also that e-cigarettes can help smokers to quit and that e-cigarettes are used only by smokers; besides the common investigation, our study also revealed that 17.6% of participants thought that using electronic cigarettes reduces the possibility for smokers to seek another medical method to quit smoking, and 46.1% thought that some teenagers use electronic cigarettes without being traditional cigarette smokers. In our study, the belief that some teenagers use e-cigarettes without being traditional cigarettes smokers was corelated with the e-cigarette experimentation reported to the whole sample and among smokers. The beliefs that e-cigarettes help quit smoking and e-cigarettes are less dangerous were also strongly correlated with e-cigarette experimentation among the whole sample, while the belief that e-cigarettes are less dangerous was correlated to experimentation with e-cigarettes also among smokers.

A study published in 2017 revealed that 77.8% of the electronic vaping product using adolescents in the study sample had tried tricks, while 83.7% had watched tricks. Performing tricks was associated with using advanced vaping devices, daily vaping, social media engagement, perceived risks, and social norms. There are few studies that mention vape tricks being a potential threat to adolescent health, and further research should examine the potential health effects of performing vape tricks [21]. The main reason for trying e-cigarettes in our study is curiosity (67.0% of the whole sample), as in the Romanian study from 2013 and in other European similar studies [22,23]. A systematic review and meta-analysis published in 2018 revealed that smoking tobacco cigarettes by family members and friends is a strong predictor of e-cigarette use in adolescence [24]. A recent study following the upward trend in e-cigarette use, using data from two Irish waves (2015, 2019) from a European School Survey on Alcohol and Other Drugs (ESPAD), revealed, besides the increasing consumption of e-cigarettes among Irish teenagers, strong correlations between friends and family influences and e-cigarette experimentation [25].

In our study, 12.9% of smokers declared that they used e-cigarettes to quit smoking. Many studies also show that the use of e-cigarettes can lead to tobacco cigarettes smoking initiation among non-smoker adolescents [26]. In our study, 3% of non-smokers declared that they will use e-cigarettes in the next year.

E-cigarettes consumption can lead to traditional smoking. A longitudinal study among Finnish adolescents that tested the association between e-cigarettes use with nicotine and nicotine-free, and daily use of tobacco based cigarettes and nicotine e-cigarettes, revealed that experimentation with nicotine e-cigarettes serves as a gateway to subsequent use of traditional cigarettes as well as nicotine e-cigarettes [27]. According to several studies, some teenagers try e-cigarettes without being tobacco cigarettes smokers [28,29]. In our study 15% of the non-smokers have tried e-cigarettes at least once during lifetime, while 4.5% have done so in the last month. Moreover, 61.7% of the smokers and 93.5% of ex-smokers declared that they were not traditional cigarettes smokers when they first tried e-cigarettes.

In the 2013 Romanian study performed among university students, 4.1% of the whole sample, and 11.7% of smokers, declared that they want to use e-cigarettes in the next year, while in a study performed on Romanian high school students in the same year, not only did the intention to use e-cigarettes increase to 13.9% for the whole sample, and to 32% for the smokers, but also 12.3% of the ex-smokers and 7.4% of the non-smokers expressed their intention to use e-cigarettes in the next year. In our study, performed among middle school aged students, the intention to use e-cigarettes in the next year was a bit lower (7.4% of the whole sample, 20.3% of smokers, 10.5% of ex-smokers, and 3.0% of non-smokers). Nevertheless, reported to the whole study sample, the intention to use e-cigarettes in the next year was stronger among those who already experimented with e-cigarettes.

According to some studies, tobacco smoking among adolescents is higher in rural areas than in urban areas [30]. A study from the USA revealed that adolescents from urban areas who were smokers were twice more likely to use e-cigarettes than adolescent smokers from rural areas [31]. These differences need to be considered in improving prevention programs concerning smoking and the use of electronic cigarettes [32].

The findings of this study reveal important factors involved in e-cigarette use initiation, but also the habits related to their use among middle school students from rural areas of Romania. These findings may lay the groundwork for educational programs to prevent the use of electronic cigarettes, considering the factors associated with initiating the use of electronic cigarettes, such as opinions and social influences, but also the novelty and characteristics of the products.

This study has several limitations. Due to logistical constraints, it included only middle school students from villages situated in rural areas from two counties of Romania, which limits the generalization of the results to the whole country. Similar to other studies in this field, the data are based on the participants’ own reports, which may cause biases; nevertheless, the study used several methods to assure participants about the confidential treatment of the provided information, hence increasing the likelihood of their honest answers.

## 5. Conclusions

E-cigarettes are a disruptive innovation raising new questions for health professionals and policymakers [3,7].

This study is one of the few studies concerning opinions and practices regarding the use of electronic cigarettes made in Romania and the first conducted among middle school students from rural areas in Romania. It confirms e-cigarette experimentation and friends and sibling’s influences, but also the association between experimentation with e-cigarettes and strong beliefs that electronic cigarettes can help smokers to quit and that e-cigarettes are less dangerous.

The increasing prevalence of smoking among middle schoolers from Romanian rural areas, but also the high prevalence of using e-cigarettes, is a good reason to implement health education programs which target both smoking prevention and the use of e-cigarettes.

## Figures and Tables

**Table 1 ijerph-19-07372-t001:** Opinions, behavior, and source of information regarding e-cigarette use, intention to use e-cigarettes, intention to quit traditional cigarettes.

	Total Sample	Smokers	Ex-Smokers	Non-Smokers
	N = 720	N = 128	N = 124	N = 468
	%	%	%	%
**Awareness**				
Ever heard about e-cigarettes	96.3	17.8	17.2	65.0
**Sources of information about e-c**				
Internet	61	53.1	62.9	62.6
Commercial/Sales points	23.5	19.5	15.3	26.7
Newspapers	3.3	4.7	0.8	3.6
Friends	42.4	60.2	48.4	35.9
People from the same school year	21.3	29.7	14.5	20.7
Parents	15.8	16.4	20.2	14.5
School Health education lessons	12.6	7.0	9.7	15.0
**Beliefs**				
**E-cigarettes are less dangerous**				
I totally agree/partially agree	37.9	53.1	47.6	31.2
I do not know	50.8	37.5	39.5	57.5
I totally disagree/partially disagree	11.3	9.4	12.9	11.3
**E-cigarettes can help smokers to quit**				
I totally agree/I partially agree	32.6	46.9	40.3	26.7
I do not know	49.9	39.1	41.1	55.1
I totally disagree/partially disagree	17.5	14.1	18.5	18.2
**E-cigarettes are used only by smokers**				
I totally agree/I partially agree	27.6	32.0	27.4	26.5
I do not know	50.4	37.5	46.0	55.1
I totally disagree/partially disagree	21.9	30.5	26.6	18.4
**Using e-c reduces the possibility for smokers to seek other medical methods to quit**				
I totally agree/I partially agree	17.6	28.1	21.8	13.7
I do not know	70.7	61.7	70.2	73.3
I totally disagree/partially disagree	11.7	10.2	8.1	13.0
**Some teens use e-c without being traditional cigarettes smokers**				
I totally agree/I partially agree	46.1	55.5	59.7	40.0
I do not know	46.8	36.7	33.9	53.0
I totally disagree/partially disagree	7.1	7.8	6.5	7.1
**Behavior**				
Used e-cigarettes at least once during the lifetime	31.7	72.7	50.8	15.4
Used e-cigarettes in the last month	7.4	20.3	4.8	4.5
Traditional cigarettes smoker when tried e-c	8.1	38.3	6.5	0
**Social influences**				
Friends have tried e-cigarettes	50.0	78.9	66.1	37.9
Mother have tried e-cigarettes	3.5	5.5	4	2.8
Father have tried e-cigarettes	4.6	3.1	4.8	4.9
Siblings have tried e-cigarettes	5.0	11.7	6.5	2.8
**Intention to use e-cigarettes in the next year**				
Definitely yes/probably yes	7.4	20.3	10.5	3.0
I do not know	14.9	36.7	21.8	7.0
Definitely no/probably no	77.8	43.0	67.7	90.0
**Intention to quit traditional cigarettes**				
Yes		74.5		
No		22.5		

The percentages are calculated for adolescents who have heard about e-cigarettes.

**Table 2 ijerph-19-07372-t002:** Reasons for trying e-cigarettes and types of e-cigarettes used.

	Total Sample	Smokers	Ex-Smokers	Non-Smokers
	N = 212	N = 95	N = 57	N = 60
	%	%	%	%
**Reasons for trying e-cigarettes** *				
E-cigarettes are less dangerous	54.7	48.4	36.8	48.3
To reduce the number of traditional cigarettes	7.1	13.7	3.5	0
To quit smoking	9.4	12.9	7.0	0
Curiosity	67.0	56.8	75.4	75.0
Other friends also tried e-cigarettes	16.5	20.0	21.1	6.7
Parents tried e-cigarettes	1.9	3.2	1.8	0
**Type of e-cigarettes used** *				
**Content in Nicotine**				
Yes	24.1	33.7	24.6	8.3
No	62.3	56.8	57.9	75.0
I do not know	22.2	24.2	21.1	20.0
**Flavors**				
Yes	85.8	90.3	84.2	81.7
No	6.6	4.3	8.8	8.3
I do not know	8.5	8.6	5.3	10.0
**Flavors** *				
Mint	26.4	30.5	26.3	20.0
Tobacco	4.2	4.2	5.3	3.3
Fruits	65.6	66.1	61.4	68.3
Chocolate	34.4	32.6	33.3	38.3
Candies	17.0	14.7	15.8	21.7
**Amount of nicotine contained in e-cigarettes** **				
1–8 mg/mL (0.1–0.8%)	19.6	18.8	21.4	20.0
9–20 mg/mL (0.9–2.0%)	15.7	15.6	14.3	20.0
>20 mg/mL (>2.0%)	5.9	6.3	0	20.0
do not know	60.8	59.4	71.4	40.0

* The percentages are calculated for middle-schoolers who ever heard about and who ever tried e-cigarettes (N = 212, smokers—95, ex-smokers—57, non-smokers—60). ** The amount of nicotine contained in e-cigarettes is referring only to middle-schoolers who declared that they had tried e-cigarettes containing nicotine (N = 51, smokers = 32; ex-smokers = 14; non-smokers = 5).

**Table 3 ijerph-19-07372-t003:** Correlates of ever trying e-cigarettes: results of univariate and multivariate logistic regression among whole sample.

	Total Sample (N = 720)			
	Univariate Logistic Regression		Multivariate Logistic Regression	
Independent Variables	Exp (B)	95% CI	Exp (B)	95% CI
**^1^ Gender**				
Female	0.788	0.575–1.079	-	-
**^2^ Age**				
14 year-old	1.514 *	1.103–2.078	0.942	0.600–1.476
**^3^ Social influences**				
Friends tried e-cigarettes				
Yes	3.835 ***	2.604–5.648	1.871 *	1.129–3.101
Mother tried e-cigarettes				
Yes	1.767	0.789–3.959	-	-
Father tried e-cigarettes				
Yes	1.874	0.925–3.794	-	-
Siblings tried e-cigarettes				
Yes	3.944 ***	1.967–7.906	2.079	0.855–5.0.54
**^4^ Beliefs**				
**E-cigarettes help quit smoking**				
I totally agree/I partially agree	4.214 ***	3.011–5.898	2.096 **	1.221–3.600
**E-cigarettes are less dangerous**				
I totally agree/I partially agree	4.173 ***	2.994–5.815	1.918 *	1.133–3.247
**E-cigarettes are used only by smokers**				
I totally agree/I partially agree	0.878	0.615–1.252	-	-
**Using e-c reduces the possibility for smokers to use another method to quit**				
I totally agree/I partially agree	2.553 ***	1.726–3.777	0.991	0.548–1.790
**Some teens use e-c without being traditional cigarettes smokers**				
I totally agree/I partially agree	2.849 ***	2.058–3.945	1.621	1.000–2.628
**^5^ Intention to use e-cigarettes in the next year**				
Definitely yes/probably yes	7.045 ***	3.739–13.274	2.482 *	1.079–5.711
**^6^ Source of information**				
**Internet**				
Yes	0.973	0.705–1.342	-	-
**Commercial**				
Yes	0.623	0.421–0.924	-	-
**Newspaper**				
Yes	0.299	0.88–1.013	-	-
**Friends**				
Yes	1.995 ***	1.451–2.743	0.938	0.579–1.520
**People from the same school year**				
Yes	0.946	0.643–1.391	-	-
**Parents**				
Yes	0.859	0.553–1.332	-	-
**School Health Education Lessons**				
Yes	0.420 **	0.239–0.738	0.532	0.242–1.167
**^7^ Smoking behavior**				
**Smoker**				
Yes	8.995 ***	5.831–13.876	6.265 ***	3.490–11.246

The reference were students who did not experiment with e-cigarettes among whole sample; ^1^ Coded as 0—male; 1—female; ^2^ Coded as 0–13 years old, 1–14 years old; ^3^ Coded as 0—no, 1—yes; ^4^ Coded as 0—totally disagree/partially disagree/do not know, 1—totally agree/partially agree; ^5^ Coded as 0—definitely no/probably no/do not know, 1—definitely yes/probably yes;^,6^ Coded as 0—no, 1—yes; ^7^ Coded as 0—non-smoker; 1—ex-smoker, 2—smoker, * *p* < 0.05, ** *p* < 0.01, *** *p* < 0.001.

**Table 4 ijerph-19-07372-t004:** Correlates of ever trying e-cigarettes: results of univariate and multivariate logistic regression among smokers.

	Smokers (N = 128)			
	Univariate Logistic Regression		Multivariate Logistic Regression	
Independent Variables	Exp (B)	95% CI	Exp (B)	95% CI
**^1^ Gender**				
Female	2.171	0.983–4.796	-	-
**^2^ Age**				
14 year-old	0.636	0.291–1.392	-	-
**^3^ Social influences**				
Friends tried e-cigarettes				
Yes	2.775 *	1.025–7.513	2.376	0.784–7.204
Mother tried e-cigarettes				
Yes	2.286	0.265–19.737	-	-
Father tried e-cigarettes				
Yes	1.620	0.174–15.060	-	-
Siblings tried e-cigarettes				
Yes	2.851	0.598–13.583	-	-
**^4^ Beliefs**				
**E-cigarettes help quit smoking**				
I totally agree/I partially agree	2.907 *	1.256–6.727	1.419	0.520–3.870
**E-cigarettes are less dangerous**				
I totally agree/I partially agree	5.013 ***	2.105–11.937	3.820 **	1.391–10.491
**E-cigarettes are used only by smokers**				
I totally agree/I partially agree	1.039	0.450–2.396	-	-
**Using e-c reduces the possibility for smokers to use another method to quit**				
I totally agree/I partially agree	1.454	0.588–3.593	-	-
**Some teens use e-c without being traditional cigarettes smokers**				
I totally agree/I partially agree	3.326 **	1.471–7.519	2.407	0.976–5.934
**^5^ Intention to use e-cigarettes in the next year**				
Definitely yes/probably yes	1.324	0.483–3.631	-	-
**^6^ Intention to quit traditional cigarettes**				
Yes	0.837	0.292–2.397	-	-
**^7^ Source of information**				
**Internet**				
Yes	0.681	0.309–1.499	-	-
**Commercial**				
Yes	1.241	0.450–3.419	-	-
**Newspaper**				
Yes	0.356	0.068–1.852	-	-
**Friends**				
Yes	1.395	0.636–3.063	-	-
**People from the same school year**				
Yes	1.077	0.457–2.537	-	-
**Parents**				
Yes	1.247	0.420–3.705	-	-
**School Health Education Lessons**				
Yes	0.270	0.068–1.070	-	-

The references were students who did not experimented with e-cigarettes among the smokers. ^1^ Coded as 0—male; 1—female; ^2^ Coded as 0–13 years old, 1–14 years old; ^3^ Coded as 0—no, 1—yes; ^4^ Coded as 0—totally disagree/partially disagree/do not know, 1—totally agree/partially agree; ^5^ Coded as 0—definitely no/probably no/do not know, 1—definitely yes/probably yes; ^6,7^ Coded as 0—no, 1—yes; * *p* < 0.05, ** *p* < 0.01, *** *p* < 0.001.

## Data Availability

The data presented is in this study are available on justified cases from the first author.

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
