# Peer review of "Opinions and Practices Regarding Electronic Cigarette Use among Middle School Students from Rural Areas of Romania"

_ijerph, 2022, doi:10.3390/ijerph19127372_

Round 1
Reviewer 1 Report
This is an interesting study on e-cigarette use in Romania. The findings are important for tobacco control researchers.
However, some changes are needed:
- Abstract: Why do the authors mention 'univariate logistic regression, rather than more precise multivariate logistic results?
- The introduction section is too long. Please limit the introduction to the most important data. Some of them may be transferred to the discussion. For example, the description of the Global Tobacco Contro and NYTS surveys can be limited to 2-3 sentences. Data on the US do not match with the EU, especially in the CEE region. Please provide a brief international context and then move to the local (Romanian) data. I fully understand the willingness to provide comprehensive background, but this change will improve the manuscript and make it easy to read and follow, with a focus on key facts.
- Methods: please provide data on the total group of schools that were used to randomly select 24 schools included in this study.
- The methods section, especially the distribution of the questionnaires is very precise which is a strong part of this study.
- Please consider adding the study questionnaire as supplementary material.
- Please provide a more logical structure of the "Data analysis" - from the most basic statistics (n; %) to more advanced methods such as logistic regression. Moreover, the following sentence should be moved to the results, rather than methods "The sample of this study is represented by middle-school students who had the consent of the parents to participate and fill in the confidential questionnaires (N=748)"
- Results: text can be limited to the most important findings. The rest of the data are easily accessible through the tables.
- This is unclear why the authors use only univariate regression rather than both, univariate and then multivariate (e.g., for those variables that were statistically significant in the univariate model).
- The discussion section is too extensive. Please focus on the most important findings.
- Conclusions should be revised. Please do not repeat the results/discussion in the conclusions. Please provide a few sentences based on your own findings. This would be a "take-home message" for the readers, rather than so extensive description that repeats facts, previously mentioned in the discussion or results section.
Author Response
Good evening. Sorry for the delay. I uploaded the response to you revision. Thank you

Round 2
Reviewer 1 Report
The Authors addressed all the comments.
Author Response
Thank you for the appreciation.
This manuscript is a resubmission of an earlier submission. The following is a list of the peer review reports and author responses from that submission.
Round 1
Reviewer 1 Report
The present study focusses on the evaluation of the knowledge, opinions and practices regarding electronic cigarettes (e-cigarettes) in Romanian adolescents from rural areas. The study includes a decent sample size and confidential questionnaires for the participants.
Authors have already published a paper "Opinions and practices regarding electronic cigarette use among Romanian high school students, Gac Sanit. Sep-Oct 2016;30(5):366-9. doi: 10.1016/j.gaceta.2016.05.001." in 2016 , which was used to evaluate the knowledge, opinions and practices regarding electronic cigarettes (e-cigarettes), and the factors associated with their use, in Romanian college students. How is this previous study different from the present one? Is the present study an extension of the previous? or its simply is an repetition?
Please improve the language too.
Author Response
Reviewer 1
1.The present study focusses on the evaluation of the knowledge, opinions and practices regarding electronic cigarettes (e-cigarettes) in Romanian adolescents from rural areas. The study includes a decent sample size and confidential questionnaires for the participants.
Authors have already published a paper "Opinions and practices regarding electronic cigarette use among Romanian high school students, Gac Sanit. Sep-Oct 2016;30(5):366-9. doi: 10.1016/j.gaceta.2016.05.001." in 2016, which was used to evaluate the knowledge, opinions and practices regarding electronic cigarettes (e-cigarettes), and the factors associated with their use, in Romanian college students. How is this previous study different from the present one? Is the present study an extension of the previous? or its simply is a repetition?
Please improve the language too.
Response:
Now in the Discussion section we indicate the novelty and importance of this study in comparison with previous study performed in Romania (please see below). The English was checked and improved.
``This study is one of the few studies concerning opinions and practices regarding the use of electronic cigarettes made in Romania. There were performed similar studies concerning the use of electronic cigarettes among Romanian university students (in 2013 and 2017)[16, 19] and among high school students in Romania (in 2013),[17] but the sample did not include middle-school aged students from rural areas from Romania. Moreover, the present study was performed in 2019, offering more recent data in comparison with previous studies; due to its changing patterns, smoking an e-cigarette use among different groups of young people should be periodically assessed. [15]’’
Reviewer 2 Report
This study assessed 1) the awareness, opinions and practices of e-cigarettes use; and 2) the factors associated with e-cigarettes use among a group of middle-school aged adolescents in rural areas of Romania. Overall, the findings from this paper are important since it may contribute to the e-cigarette use literature especially in rural Romania, but many things need to think about and be improved.
- There are many typos or punctuation mark mistakes throughout the manuscript. It seems like authors were in a rush to finish and submit this manuscript in the past. For example, for abstract, in methods, “…related to the use of electronic were performed…” should be “…related to the use of e-cigarettes was performed…”; 4.5%; 50.8%; Global Tobacco Control 2020, etc.
- Authors should make electronic cigarette or e-cigarette consistent in the manuscript. I suggest author could define e-cigarette as an abbreviation of electronic cigarette. For example, use electronic cigarette (e-cigarette) at the beginning and only use e-cigarette in the following.
- For introduction, here is a recent citation that authors may want to use instead of using citation [4].
Sapru, S., Vardhan, M., Li, Q., Guo, Y., Li, X., Saxena, D., 2020. E-cigarettes use in the United States: reasons for use, perceptions, and effects on health. BMC Public Health. 20(1), 1518. https://doi.org/10.1186/s12889-020-09572-x.
- Introduction should be expanded more regarding the adverse outcomes from using e-cigarettes such as smoking initiation though authors mentioned it in the discussion part. In addition, authors should state the rational of why they conducted this study in rural areas only. There is lack of the location background description.
- As this study recruited adolescents aged 13-14 years old, I would suggest that authors use middle school aged description or change children, teenagers, or adolescents into middle schoolers. Therefore, the population involved in this study is more straightforward and specified.
- Is there any chance that the middle schoolers will underreport their use of e-cigarettes? How will authors deal with the potential bias?
- For methods, the paragraph started with “the field of research regarding e-cigarette use has only emerged….” should go to the introduction.
- Social influence appeared abruptly in the results since it was not described/defined in the methods.
- The results are too much, and it is hard to follow, suggesting that using subtitles or describing by smokers, ex-smokers, non-smokers. Need to work more on those.
- Some of the results make no sense to me. For example, in table 2, how to explain 5.6% of non-smokers trying e-cigarettes for the reason to quit smoking? They are not smoking, right?
- Some content in the first paragraph of the discussion should go to introduction.
- Table 3 is too long, suggesting removing the reference groups and only leaving the comparative groups to save some space and easier to read.
- For the discussion, I would suggest authors discussing the debate on e-cigarette use (smoking cessation for adults and may lead to e-cigarette initiation among youth who do not use tobacco). Expand this based on the findings will be more interesting and useful.
- Last, I would like to recommend authors to ask some official language companies to help them with their sentences writing.
Author Response
Reviwer 2
This study assessed 1) the awareness, opinions and practices of e-cigarettes use; and 2) the factors associated with e-cigarettes use among a group of middle-school aged adolescents in rural areas of Romania. Overall, the findings from this paper are important since it may contribute to the e-cigarette use literature especially in rural Romania, but many things need to think about and be improved.
- There are many typos or punctuation mark mistakes throughout the manuscript. It seems like authors were in a rush to finish and submit this manuscript in the past. For example, for abstract, in methods, “…related to the use of electronic were performed…” should be “…related to the use of e-cigarettes was performed…”; 4.5%; 50.8%; Global Tobacco Control 2020, etc.
Response
We corrected the typo and punctuation mark mistakes. Hopefully we haven`t miss anything.
- Authors should make electronic cigarette or e-cigarette consistent in the manuscript. I suggest author could define e-cigarette as an abbreviation of electronic cigarette. For example, use electronic cigarette (e-cigarette) at the beginning and only use e-cigarette in the following.
Response:
As you suggested, we used electronic cigarette(e-cigarette) at the beginning, and e- cigarette in the following parts.
- For introduction, here is a recent citation that authors may want to use instead of using citation [4].
Sapru, S., Vardhan, M., Li, Q., Guo, Y., Li, X., Saxena, D., 2020. E-cigarettes use in the United States: reasons for use, perceptions, and effects on health. BMC Public Health. 20(1), 1518. https://doi.org/10.1186/s12889-020-09572-x.
Response: We kept the old citation, but we also added the suggested citation, which indeed is very relevant.
- Introduction should be expanded more regarding the adverse outcomes from using e-cigarettes such as smoking initiation though authors mentioned it in the discussion part. In addition, authors should state the rational of why they conducted this study in rural areas only. There is lack of the location background description.
Response:
-In the Introduction we include more information regarding the adverse outcomes of e-cigarettes (please see below)
‘’The particularities of an adolescent’s developing brain and his vulnerability to negative effects of nicotine neurotoxin and nicotine dependence, make cigarettes consumption in adolescence, even more dangerous.[10] Oxidative stress is a key molecular factor that drives the harmful effects of traditional cigarettes smoking. E-cigarettes, just like traditional ones, induces oxidative stress, which has harmful effects on the developing brain of adolescents; the oxidative stress appears due to many e-cigarettes’ components, including the flavours, vapor, e-liquids. Due to oxidative stress, e-cigarettes use could lead to poor learning and academic performance, aggressive and impulsive behaviour, poor sleep quality, impaired memory and cognition, attention deficits, but also, can increase the risk of depression and suicidal ideation. [11]
A meta-analysis published in 2021 showed that the use of e-cigarettes in adolescents may be associated with smoking initiation.[12] The World Health Organization underlines that tobacco products and e-cigarettes pose risks to health, and the safest approach is not to consume.[13]”
- In the introduction we now better explain the rationale for developing the study in rural areas (please see below), while in the limitations part we acknowledge the fact that this is not a national representative study
A meta-analysis published in 2021 showed that the use of e-cigarettes in adolescents may be associated with smoking initiation.[12] The World Health Organization underlines that tobacco products and e-cigarettes pose risks to health, and the safest approach is not to consume.[13]
The field of research regarding e-cigarette use has only emerged in the last few years, and no standardized questionnaire is available, with several international studies trying to assess both behaviours as well as knowledge, attitudes, reasons related to trying/continuing e-cigarette use, or intention to use them in the future. [3–6]
According to some studies, tobacco smoking among adolescents is higher in rural areas than in urban areas.[7] A study from the USA revealed that adolescents from urban areas who were smokers were twice more likely to use electronic cigarettes than adolescent smokers from rural areas.[8] These differences are interesting and they need to be considered in improving prevention programs concerning smoking and the use of electronic cigarettes.[9]
Because of these findings, according to Global Tobacco Control 2020, e-cigarettes have been banned in 41 countries; 66 countries allow the sale of e-cigarettes, but provide restrictions/regulations on sale; from these 66, at least 32 have regulations on the concentration and volume of nicotine in e-liquids, and they do not allow the use of ingredients that pose a risk to human health under heated or unheated form in the e-liquid containing nicotine.[13–15]
Even though there are many countries where e-cigarettes are banned, in several European countries, such as Romania, e-cigarettes are legal and highly promoted. Given the addictive and harmful effects of e-cigarettes, it is essential to identify the factors associated with the initiation and use of e-cigarettes among young people in order to develop appropriate health education programs and policy measures. Nevertheless, in Romania this type of information is limited, while no information is available with regard to e-cigarette use among adolescents from rural areas.
This study aims to assess awareness, opinions, and practices regarding e-cigarette use among Romanian middle-schoolers from rural areas and to identify correlates of experimentation with e-cigarettes among them.
- There is lack of the location background description.
Response
We included more information about this issue (please see below)
‘’ The schools involved in the study were situated in rural areas from two counties of Romania; one situated in North-western part of the country (Cluj County) and one from the west side of Romania (Arad County).
Principals of twenty-four school randomly selected from rural areas situated in the selected counties were asked if they agree with the participation of their school in a smoking prevention program.
The schools were randomly selected from the list of the gymnasium schools from villages located on the selected rural areas. Twenty-two out of twenty-four school principals agreed to participate with their schools in the project and provided the number of the 7th and 8th-grade classes that could participate.
Informed consent of parents for the participation of middle school students in the study was also looked for. A number of 825 parents, out of 1172 that were asked to offer informed consent for students’ participation, approved their children's participation in the study. ‘’
- As this study recruited adolescents aged 13-14 years old, I would suggest that authors use middle school aged description or change children, teenagers, or adolescents into middle schoolers. Therefore, the population involved in this study is more straightforward and specified.
Answer: Thank you for the suggestion. As you suggested, we changed children, teenagers or adolescents into middle schoolers.
- Is there any chance that the middle schoolers will underreport their use of e-cigarettes? How will authors deal with the potential bias?
Answer:
In the Methodology we describe in detail the procedure used to assure trust of participants that the date will be treated confidentially, but in the Limitations part we acknowledge that possible biases might appear (please see below)
‘’ This study has several limitations. Due to logistical constraints, it included only middle school students from villages situated in rural areas from two counties of Romania, which limits the generalization of the results to the whole country. Also, similar with other studies in this field, the data are based on participant’s own reports, which may cause biases; nevertheless, the study used several methods to assure participants about the confidential treatment of the provided information, hence increasing the likelihood of their honest answers. ‘’
- For methods, the paragraph started with “the field of research regarding e-cigarette use has only emerged….” should go to the introduction.
Answer: We added to the introduction the paragraph that you suggested.
- Social influence appeared abruptly in the results since it was not described/defined in the methods.
We included this information in the description of the questionnaire (please see below)
‘’ Social influences related to e-cigarette use (middle schoolers were asked if their parents/siblings/friends are using e-cigarettes-the possibilities of answer were yes/no/I do not know; for the question concerning the parents and siblings, they had also the possibility to answer,, I do not have brother/sister’’ or ,, my mother/father is dead’’
- The results are too much, and it is hard to follow, suggesting that using subtitles or describing by smokers, ex-smokers, non-smokers. Need to work more on those.
We have tried to focus on smokers, ex-smokers, non-smokers
- Some of the results make no sense to me. For example, in table 2, how to explain 5.6% of non-smokers trying e-cigarettes for the reason to quit smoking? They are not smoking, right?
Answer:
This was a transcription error, sorry.
- Some content in the first paragraph of the discussion should go to introduction.
Answer
We reorganised both the introduction and discussion section
-Table 3 is too long, suggesting removing the reference groups and only leaving the comparative groups to save some space and easier to read.
Answer:
We made this change
- For the discussion, I would suggest authors discussing the debate on e-cigarette use (smoking cessation for adults and may lead to e-cigarette initiation among youth who do not use tobacco). Expand this based on the findings will be more interesting and useful.
Answer
We incorporated this comment in the new version (please see below)
The main reason for trying e-cigarettes in our study is curiosity (67.0% of the whole sample), as in the Romanian study from 2013 and in other European similar studies, followed by the belief that electronic cigarettes are less dangerous than traditional cigarettes (54.7% of the whole sample), and the fact that friends use them too (16.5% of the whole sample).[4,26] A systematic review and meta-analysis published in 2018, revealed that smoking tobacco cigarettes by family members and friends is a strong predictor of e-cigarettes use in adolescence.[27] A recent study following the upward trend in e-cigarette use, using data from two Irish waves (2015,2019) from an European School Survey on Alcohol and Other Drugs (ESPAD), revealed, besides the increasing consumption of e-cigarettes among Irish teenagers, strong correlations between friends and family influences and e-cigarettes experimentation. [28]
A meta-analysis published in February 2021, revealed that in observational studies, e-cigarettes were not associated with increased smoking cessation in the adult population. In our study, 12.9% of smokers declared that they used e-cigarettes to quit smoking. Also, many studies show that the use of e-cigarettes can lead to e-cigarettes smoking initiation among adolescents who don`t smoke tobacco cigarettes. In our study, 3% of non-smokers, declared that they will use e-cigarettes in the next year.
E-cigarettes consumption can lead to traditional smoking. A longitudinal study among Finnish adolescents, who tested the association between e-cigarettes use, with nicotine and nicotine-free, and daily use of tobacco based cigarettes and nicotine e-cigarettes, revealed that experimentation with nicotine e-cigarettes serves as a gateway to subsequent use of traditional cigarettes as well as nicotine e-cigarettes.[29] According to several studies, some teenagers try e-cigarettes, without being tobacco cigarettes smokers.[30,31] In our study 15% of the non-smokers have tried e-cigarettes at least once during lifetime, while 4.5% have done so in the last month. Moreover, 61.7% of the smokers, 93.5% of ex-smokers declared that they were not traditional cigarettes smokers when they first tried e-cigarettes.
- Last, I would like to recommend authors to ask some official language companies to help them with their sentences writing.
Answer
English was checked and corrected.
Reviewer 3:
- Please clearly justify the representativeness of the study sample.
We included in the introduction information about the procedure for data collection and the characteristics of the study (please see below), while in the limitations part we acknowledge the fact that due to logistical constraints, it included only middle school students from villages situated in rural areas from two counties of Romania, which limits the generalization of the results to the whole country
The schools involved in the study were situated in rural areas from two counties of Romania; one situated in North-western part of the country (Cluj County) and one from the west side of Romania (Arad County).
Principals of twenty-four school randomly selected from rural areas situated in the selected counties were asked if they agree with the participation of their school in a smoking prevention program.
The schools were randomly selected from the list of the gymnasium schools from villages located on the selected rural areas. Twenty-two out of twenty-four school principals agreed to participate with their schools in the project and provided the number of the 7th and 8th-grade classes that could participate.
Informed consent of parents for the participation of middle school students in the study was also looked for. A number of 825 parents, out of 1172 that were asked to offer informed consent for students’ participation, approved their children's participation in the study.
- Table 1 and 2: Please remove "%" from the second line where the "N" is reported
We included the % on the next line
Reviewer 3 Report
This is a well-prepared manuscript. This study aims to assess awareness, opinions, and practices regarding electronic cigarette use among gymnasium students from Romania, and to identify correlates of experimentation with e-cigarettes among Romanian adolescents.
The study aim is in line with the IJERPH objectives, but some changes are needed:
1) Please clearly justify the representativeness of the study sample.
2) Table 1 and 2: Please remove "%" from the second line where the "N" is reported
3) Please clearly define the practical implications of this study
4) Please clearly define the limitations of this study
Author Response
- Please clearly justify the representativeness of the study sample.
We included in the introduction information about the procedure for data collection and the characteristics of the study (please see below), while in the limitations part we acknowledge the fact that due to logistical constraints, it included only middle school students from villages situated in rural areas from two counties of Romania, which limits the generalization of the results to the whole country
The schools involved in the study were situated in rural areas from two counties of Romania; one situated in North-western part of the country (Cluj County) and one from the west side of Romania (Arad County).
Principals of twenty-four school randomly selected from rural areas situated in the selected counties were asked if they agree with the participation of their school in a smoking prevention program.
The schools were randomly selected from the list of the gymnasium schools from villages located on the selected rural areas. Twenty-two out of twenty-four school principals agreed to participate with their schools in the project and provided the number of the 7th and 8th-grade classes that could participate.
Informed consent of parents for the participation of middle school students in the study was also looked for. A number of 825 parents, out of 1172 that were asked to offer informed consent for students’ participation, approved their children's participation in the study.
- Table 1 and 2: Please remove "%" from the second line where the "N" is reported
We included the % on the next line
- Please clearly define the practical implications of this study
In the Conclusions section we expand the practical implications of the study (please see below)
This study is one of the few studies concerning opinions and practices regarding the use of electronic cigarettes made in Romania and the first conducted among middle school students from rural areas in Romania. It confirms the association between using e-cigarettes experimentation and friends and sibling’s influences, but also the association between experimentation with electronic cigarettes and strong beliefs that electronic cigarettes can help smokers to quit and that e-cigarettes are less dangerous.
The prevalence of smoking among middle schoolers from Romanian rural areas, but also the prevalence of using electronic cigarettes, is a good reason to implement health education programs which targets both smoking prevention, but also the use of electronic cigarettes
The findings of this study reveal important factors involved in e-cigarette use initiation, but also the habits related to their use among middle school aged students from rural areas of Romania. These findings may lay the groundwork for educational programs to prevent the use of electronic cigarettes, considering the factors associated with initiating the use of electronic cigarettes, such as opinions and social influences, but also the novelty and characteristics of the products.
The results imply that educational programs regarding prevention of electronic cigarette use should offer clear and trusted information about the consequences of their use, should try to help middle-schoolers to identify misperceptions promoted in different ways, the influences coming from peers, but also promotion tactics, in order to develop skills as well as action plans to cope with pressure to use electronic cigarettes ; these findings are similar to those from other international studies.[6,14,16,18,31] Moreover, it shows once again, the importance to embrace health education programs in broader tobacco control policies and measures, including those targeting the novel products, in order to decrease the availability and social acceptability of these products and to help children and their parents to adopt a lifestyle free of smoking and use of other novel products, too.
The findings of this study reveal important factors involved in e-cigarette use initiation, but also the habits related to their use among middle school aged students from rural areas of Romania. These findings may lay the groundwork for educational programs to prevent the use of electronic cigarettes, considering the factors associated with initiating the use of electronic cigarettes, such as opinions and social influences, but also the novelty and characteristics of the products.
The results imply that educational programs regarding prevention of electronic cigarette use should offer clear and trusted information about the consequences of their use, should try to help middle-schoolers to identify misperceptions promoted in different ways, the influences coming from peers, but also promotion tactics, in order to develop skills as well as action plans to cope with pressure to use electronic cigarettes ; these findings are similar to those from other international studies.[6,14,16,18,31] Moreover, it shows once again, the importance to embrace health education programs in broader tobacco control policies and measures, including those targeting the novel products, in order to decrease the availability and social acceptability of these products and to help children and their parents to adopt a lifestyle free of smoking and use of other novel products, too.
- Please clearly define the limitations of this study
We included this section in the revised version (please see below)
This study has several limitations. Due to logistical constraints, it included only middle school students from villages situated in rural areas from two counties of Romania, which limits the generalization of the results to the whole country. Also, similar with other studies in this field, the data are based on participant’s own reports, which may cause biases; nevertheless, the study used several methods to assure participants about the confidential treatment of the provided information, hence increasing the likelihood of their honest answers.
Round 2
Reviewer 2 Report
I was a reviewer on the original paper. Thanks to the authors for studiously modifying the paper. I believe it is much improved and I am satisfied with the revision.
Only one comment, please pay attention to the typo mistakes throughout the manuscript, such as punctuations and spaces.